# The Stability of Social and Behavioral Rhythms and Unexpected Low Rate of Relevant Depressive Symptoms in Old Adults during the COVID-19 Pandemic

**DOI:** 10.3390/jcm13072005

**Published:** 2024-03-29

**Authors:** Federica Sancassiani, Giulia Cossu, Elisa Cantone, Ferdinando Romano, Alessandra Perra, Antonio Urban, Samantha Pinna, Stefano Del Giacco, Roberto Littera, Davide Firinu, Luchino Chessa, Enzo Tramontano, Antonio Egidio Nardi, Mauro Giovanni Carta

**Affiliations:** 1Department of Medical Sciences and Public Health, University of Cagliari, 09124 Cagliari, Italy; giuliaci@hotmail.com (G.C.); elisa.cantone@libero.it (E.C.); alessandra.perra@unica.it (A.P.); a.urban@aoucagliari.it (A.U.); samanthapinna1984@gmail.com (S.P.); delgiac@gmail.com (S.D.G.); davidefirinu@yahoo.it (D.F.); luchinochessa@unica.it (L.C.); maurogcarta@gmail.com (M.G.C.); 2Department of Public Health and Infectious Diseases, University of Rome “La Sapienza”, 00185 Roma, Italy; ferdinando.romano@uniroma1.it; 3University Hospital of Cagliari, 09124 Cagliari, Italy; 4Medical Genetics, “R. Binaghi” Hospital, 09126 Cagliari, Italy; roby.litter@gmail.com; 5AART-ODV (Association for the Advancement of Research on Transplantation), 09131 Cagliari, Italy; 6Department of Life and Environmental Sciences, University of Cagliari, 09124 Cagliari, Italy; tramon@unica.it; 7Institute of Psychiatry-IPUB, Federal University of Rio de Janeiro, Rio de Janeiro 22290-140, Brazil; antonioenardi@gmail.com

**Keywords:** COVID-19, pandemic, elderly, depression, circadian rhythms, social rhythms, light pollution

## Abstract

**Background**: The disruption of social rhythms was found to be associated with depressive disorders during the COVID-19 pandemic; lower rates of these disorders were surprisingly found in old adults. The present study aims to verify the stability of social rhythms during lockdown in a sample of elderly people. **Methods**: Controlled cohort study (secondary analyses) of a previous randomized-controlled trial with the first evaluation in April 2019 (T0) and then 48 weeks later (T1) during the lockdown. The regulation of social and behavioral rhythms was measured through the Brief Social Rhythms Scale (BSRS); the Patient Health Questionnaire-9 (PHQ9) was adopted to detect relevant depressive symptoms. **Results**: 93 elderlies (73.36 ± 4.97 years old, 50.5% females) were evaluated at T0 and T1. Neither the total score of BSRS nor any of the 10 items showed a statistically significant difference comparing the two survey periods. The frequency of relevant depressive symptoms was 5.3% at T0 and 6.4% at T1 (OR = 0.8, CI95% 0.2–24). **Conclusions**: Among elderlies who did not show an increased risk of depression during the lockdown, social and behavioral rhythms remained exceptionally stable during the same period. Considering previous evidence about rhythms dysregulation preceding depression, their stability may be considered a factor of resilience.

## 1. Introduction

Mood disorders are closely associated with the dysregulation of biological and social rhythms (sleeping, eating, carrying out social activities) which are, in turn, directly related to circadian biorhythms [1,2]. The disrupted social and circadian rhythms resulting from the stringent lockdown measures have been scrutinized, among other factors, as potential explanations for the heightened prevalence of depressive disorders induced by the COVID-19 pandemic [3,4].

In a community study conducted in France, a delay in chronotype was identified in association with depressive symptoms during lockdowns [5]. In a comparative study between two climatically similar cities with contrasting degrees of lockdown stringency imposed by authorities, individuals diagnosed with bipolar disorder in Cagliari (Italy) were identified as having a higher risk of experiencing depressive relapses compared to their counterparts with the same diagnosis in Tunis (Tunisia) during the initial wave of the COVID-19 pandemic [6].

The occurrence of depression, linked with the notably stringent lockdown imposed by Italian authorities, was noted to be associated with the disruption of social rhythms, encompassing alterations in sleep patterns, eating habits, and social interactions [6]. The exploration of potential specific resilience mechanisms was undertaken in a follow-up of a randomized-controlled trial (RCT) on exercise started in 2019 [7,8]. The study protocol included a telephone follow-up conducted 48 weeks after the onset of the trial, serendipitously aligning with the lockdown [9]. In the group of elderly people who participated in the trial, it was found that the good regulation of social rhythms one year before the COVID-19 pandemic was associated with a low risk of depression one year later, during the pandemic [9]. This association was determined on the premise that rhythm dysregulation may act as a risk factor for depression [9]. It has been hypothesized that the syndrome of dysregulation of social rhythms (DYMERS) is a condition linked to distress [10] and a possible common vulnerability triggers different disorders due to specific individual risk factors, such as the genetic aptitude for hyperactivity in some mood disorders [11,12,13].

Light serves as the primary environmental cue affecting circadian biorhythms and, consequently, social rhythms [14]. During periods when individuals have limited exposure to outdoor light, such as in a lockdown, indoor light assumes a predominant role in synchronizing the 24-h biorhythm [15,16]. Artificial illumination has the potential to disrupt the timing of melatonin incretion and, consequently, impact sleep [17]. Evidence also suggests age-related variations in light exposure patterns among individuals residing in the community [18,19], and a study conducted a few years ago discovered that healthy older individuals were subjected to significantly higher levels of external light compared to their younger counterparts [20]. The hindrance to leaving the residence during the lockdown period should, therefore, have led to a more pronounced dysregulation of rhythms among the elderly.

During the COVID-19 pandemic, older adults were more prone than other age groups to experience restricted social interactions with family, friends, and stakeholders, disruptions in daily activities, limited access to healthcare, and challenges in adapting to remote care such as telemedicine [21]. The elderly also faced the psychological consequences of being cognizant that their age put them at a higher risk of mortality. Indeed, during the initial waves of the COVID-19 pandemic, older adults exhibited higher mortality rates compared to other age groups [22]. However, different surveys carried out in high-income countries surprisingly found lower rates of distress and depressive disorders in old adults living at home, compared to other age groups [23,24,25,26].

The present study aims to evaluate, by secondary analyses, the stability of social rhythms among the elderly population assessed in the aforementioned RCT [7,8] during the COVID-19 pandemic. This involves comparing the initial assessment (T0) with a follow-up conducted one year later (T1) during the lockdown. Therefore, considering our previous investigation that identified rhythm dysregulation as a determinant of depression [9], the stability of social and behavioral rhythms among the elderly may be regarded as a specific facet of resilience.

## 2. Material and Methods

### 2.1. Design and Sample

A controlled cohort study (secondary analyses) was carried out in Cagliari, Italy on elderly people previously involved in an RCT on exercise for active aging [7,8] and on elderly people recruited for the same RCT, but not enrolled in the study arms due to overnumbering, and remained in a waiting list for another trial.

As reported previously [7,8], the inclusion criteria were: people living at home, 65 years old and older, both genders, holding a medical certificate for non-competitive physical activity. Exclusion criteria were: age ≤ 65 years old, BMI > 35, unsuitability for moderate physical activity due to any medical condition, lifetime history of psychosis and/or mania, a certified organic brain disease, already involved in a program of physical exercise at a level equal or higher than 70% of the expected physical activity level of the intervention study.

All people were evaluated for depressive symptoms and rhythms dysfunction at the cohort entry in April 2019 (T0) and 48 weeks after T0, in April 2020 (T1). To assess the association between dysfunctional rhythms and depressive symptoms, the entire cohort was divided into individuals with rhythms dysfunction (BSRS score: above the mean plus one standard deviation) and individuals without rhythms dysfunction, at T0 and T1.

### 2.2. Tools

The regularity and functionality of social and behavioral rhythms were measured through the Brief Social Rhythms Scale (BSRS) [27] in the Italian version [28]. This scale consists of ten items that measure the variability in daily routines across the workweek and weekend. Specifically, it assesses the regularity of sleep patterns, mealtimes, and social interactions. The higher the score, the worse the functioning and regularity of social and individual rhythms. Cronbach’s alpha coefficient was 0.912 [28]. Without a standardized cut-off, we used scores above the mean plus one standard deviation to indicate rhythm dysfunction.

The Patient Health Questionnaire-9 (PHQ9) [29], a self-administered tool, in the Italian version [30], was adopted to detect Relevant Depressive Symptoms (RDS). The nine items reflect the core symptoms of the Diagnostic and Statistical Manual of Mental Disorders (DSM) criteria for a Depressive Episode. Cronbach’s alpha coefficient was 0.79 [29]. In this study, people with a PHQ9 score > 7 were considered to have RDS and a high probability of a Depressive Episode.

A copy of the Italian version of both questionnaires (PHQ-9 and BSRS) is included in the Appendix A.

### 2.3. Statistical Analysis

The data were analyzed using the software SPSS (version 23). The differences from T0 to T1 regarding the total score and each item of BSRS in the whole sample were measured using the ANOVA 1-way for repeated measures.

The differences from T0 to T1 regarding people with relevant depressive symptoms and/or rhythms dysregulation were measured using the Chi-Square test with Yates correction.

Through the Chi-square test with Yates correction, the homogeneity between groups (with vs. without dysfunctional rhythms) has been verified at T0 regarding the variable “gender”.

The Fisher exact test was used to compare: (1) people with Relevant Depressive Symptoms plus Dysfunctional Rhythms and the remaining portions of the cohort at T0 and T1 and (2) new cases with Relevant Depressive Symptoms at T1 with increasing >9 points in BSRS total score from T0 to T1, and the remaining portions of the cohort.

All the statistics were considered as significant if *p* < 0.05.

### 2.4. Ethics

The study was carried out in adherence to the principles of the Declaration of Helsinki. The Ethics Committee of the University Hospital of Cagliari approved the study on 25 October 2018 (registered with the number: PG/2018/15546). Written informed consent was obtained from all the subjects involved in the study.

## 3. Results

Ninety-three elderly individuals (mean ± SD age: 73.36 ± 4.97 years; 47 females [50.5%]) underwent assessment 48 weeks (T1) after the initial evaluation (T0) conducted for recruitment in a previous randomized controlled trial (RCT) [7,8]. The total sample (N = 93) included: 44 people enrolled in the exercise arm of the previous RCT (mean ± SD age: 73.0 ± 5.07; 22 females [50.0%]), 39 people enrolled in the cultural activities arm in the previous RCT (mean ± SD age 73.82 ± 4.75; 21 females [53.8%]), 10 people recruited for the same RCT, not enrolled in the study arms due to the overnumbered, that remained in the waiting list for another trial, but evaluated both at T0 and T1 (mean ± SD age: 72.90 ± 5.20; four females [40.0%]).

Table 1 shows comparisons regarding the total score and each item of the BSRS from the first assessment (T0) to 48 weeks after (T1), that is, during the lockdown in the entire sample. Neither the total score nor any of the 10 items show a statistically significant difference.

Comparing T0 and T1, there was not any statistically significant difference regarding the frequency of people with dysfunctional rhythms (32.2% vs. 24.7%, OR = 1.45, CI95% 0.76–2.75), the frequency of people with relevant depressive symptoms (5.4% vs. 6.4%, OR = 0.82, CI95% 0.24–2.8) and the frequency of people with relevant depressive symptoms plus dysfunctional rhythms (80% vs. 66.7%, OR = 2, CI95% 0.13–31.98) (see Table 2).

At T0, among the 30 subjects with dysfunctional rhythms, 10 were males, balanced with the 36 males among the 63 subjects without dysfunctional rhythms (χ^2^ with Yates correction = 1.1879, *p* = 0.275).

Among people with relevant depressive symptoms, there was a higher frequency of people with dysfunctional rhythms than the remaining portions of the cohort, both at T0 (*p* = 0.036) and T1 (*p* = 0.0313) (see Table 3).

Finally, three people were found as new cases with Relevant Depressive Symptoms (RDS) at T1, three RDS cases were already present at T0, and 2 RDS cases at T0 had recovered during the follow-up at T1. Table 4 shows that the frequency of people with an increase > 9 points in the BSRS total score among the three new RDS cases at T1 was 66.6%, against 5.6% in the remaining portions of the cohort (Fisher Exact test *p* = 0.014, OR = 34.4 2.6–441.7).

## 4. Discussion

Consistent with previous research [23,24,25,26], the current study demonstrated that a cohort of elderly individuals evaluated before and after 48 weeks throughout the initial lockdown prompted by the COVID-19 pandemic did not exhibit a rise in depressive symptoms. The study additionally revealed that the same sample of elderly individuals exhibited no alterations in the dysregulation of behavioral and social rhythms during lockdown compared to the previous evaluation.

These data, specifically conducted in older adults, diverge from observations in at-risk populations of adults and young adults during the COVID-19 pandemic, where the dysregulation of behavioral rhythms, particularly in sleep, was noted in association with depressive symptoms [5]. A previous study conducted by our research group emphasized that individuals with bipolar disorder undergoing a stringent lockdown exhibited higher rates of depressive relapses compared to a similar control sample from a city where a less restrictive lockdown was implemented [6]. The increased prevalence of depressive relapses was associated with dysregulation in social and behavioral rhythms, including disruptions in sleep, eating patterns, and social interactions [6].

Although suggesting only a trend, the findings of the current study further substantiate the association between the exacerbation of rhythm dysregulation and the manifestation of depressive symptoms. Specifically, even if small, the subset of individuals with relevant depressive symptoms (RDS) exhibited a substantial dysregulation in social and behavioral rhythms. Moreover, individuals detected as new cases with RDS which emerged during the lockdown, showed an increase in their BSRS score by >9 points 34 times more than individuals not detected as new cases with RDS in the same period.

The rise in depressive symptoms attributed to sleep dysregulation during the lockdown prompted by the COVID-19 pandemic has also been proposed as a potential mediator for the documented upsurge in eating disorders during that timeframe [31]. Nevertheless, these articles failed to elucidate the question regarding the pathogenetic direction of the association, namely whether dysregulation preceded depression or vice-versa [5]. An initial study extracted from our database appeared to be a pivotal contribution to this inquiry, as it revealed that the dysregulation of rhythms 48 weeks before lockdown served as a risk factor for depression during lockdown [9]. The current study underscored that within a sample of elderly individuals who did not exhibit an increased risk of depression during lockdown, social and behavioral rhythms remained exceptionally stable throughout the same period. In consideration of prior evidence, this may imply that the stability of social and behavioral rhythms could be a factor of resilience. However, further investigation is warranted to comprehend the reasons behind the sustained stability of social rhythms in the elderly, contrasting with other age groups, during this specific period.

A recent study, employing objective telemetric measures, documented a remarkable increase in the utilization of video games during the lockdown periods of the COVID-19 pandemic [32]. During the same periods, there was a rise in distance learning [33] and remote working [34,35]. Engaging in these activities exposes individuals to significant light pollution, particularly with the use of e-tablets, a practice less common among older adults [36]. It is widely recognized that the excessive use of computers and, to an even greater extent, e-tablets, exposes individuals to the risk of light pollution [37]. In the altered conditions brought about by the lockdown, it is plausible that elderly adults may experience reduced exposure to light pollution, a factor intricately linked to the regulation of biological rhythms through melatonin. This could be attributed to their retirement status, which may imply less engagement in remote work, or cultural preferences leading to less use of e-tablets and computers.

Other factors may be considered or implicated as contributing causes to the emergence of the paradox wherein the rate of depressive symptoms did not increase during the lockdown in the elderly.

In our study, the first and the follow-up evaluations were performed in April 2019 and April 2020, respectively. In Cagliari (Italy), there were approximately 13 h of daylight at this time [38], which is slightly above the annual average. For individuals with work and school obligations, the lockdown might have posed more challenges in terms of adaptation compared to retired elderly individuals whose days were characterized by fewer hectic extra-domestic commitments. Moreover, in the countries examined in the research regarding the reduced risk of depression in the elderly during the COVID-19 pandemic (USA, Canada, Italy), the majority of older individuals received pension benefits. Unlike workers in commercial or business sectors whose incomes were at risk, the elderly were less exposed to economic uncertainty during the pandemic. Hence, it is plausible that the elderly faced less economic uncertainty. Nevertheless, the unexpected stability of social and behavioral rhythms identified by our study among the elderly during the COVID-19 pandemic has proven to be closely related to the low risk of depression in old adults, as later confirmed by several studies. Therefore, the data need validation and additional exploration, as does the hypothesis of a central role in lower exposure to light pollution.

The investigation exhibits some limitations. It was carried out with small and unstructured samples, and the high variability in the data could play a role in undermining the statistical power of the findings. Furthermore, assessing only for social/behavioral rhythms without accounting for potential confounding variables (i.e., exposure to light intensity, both indoors and outdoors; comorbid somatic and/or psychiatric conditions; concomitant pharmacological treatments; lifestyle factors, hours/day spent interacting with electronic devices), precludes any definitive conclusions and does not robustly verify the hypothesis regarding the stability of rhythms, as well as the association between rhythm dysregulation and relevant depressive symptoms.

## 5. Conclusions

Among a cohort of elderly people who did not show an increased risk of depression during the lockdown prompted by the COVID-19 pandemic, social and behavioral rhythms remained exceptionally stable during the same period. Considering previous evidence about rhythms dysregulation preceding depression, their stability may be considered a factor of resilience in this population.

Notably, the unprecedented conditions of the COVID-19 pandemic allowed us just to serendipitously test our hypothesis through a longitudinal study that pointed out interesting results, even if preliminary. They opened the way to the formulation of novel research hypotheses (i.e., investigating the role of exposure to light pollution in the regulation of social and behavioral rhythms among the elderly) to be further explored by randomized-controlled studies with larger samples.

## Figures and Tables

**Table 1 jcm-13-02005-t001:** Change from T0 to T1 regarding social and behavioral rhythms in the entire sample.

	BSRS (T0) Mean ± sd	BSRS (T1) (during Lockdown) Mean ± sd	1-Way ANOVA df 1.84
BSRS total score	20.21 ± 8.13	20.90 ± 8.82	F = 1.860; *p* = 0.74
BSRS item 10	1.67 ± 0.91	1.74 ± 0.90	F = 0.278; *p* = 0.599
BSRS item 9	1.51 ± 0.86	1.54 ± 0.68	F = 0.070; *p* = 0.792
BSRS item 8	2.32 ± 1.44	2.07 ± 1.14	F = 2.019; *p* = 0.158
BSRS item 7	2.26 ± 1.39	2.15 ± 1.15	F = 0.346; *p* = 0.557
BSRS item 6	2.32 ± 1.36	2.26 ± 1.23	F = 1.00; *p* = 0.753
BSRS item 5	2.24 ± 1.34	2.04 ± 1.12	F = 1.220; *p* = 0.271
BSRS item 4	2.04 ± 1.14	1.88 ± 0.91	F = 1.138; *p* = 0.287
BSRS item 3	1.91 ± 1.01	1.79 ± 0.85	F = 0.769; *p* = 0.382
BSRS item 2	2.04 ± 1.03	2.05 ± 0.99	F = 0.005; *p* = 0.946
BSRS item 1	1.86 ± 0.93	1.85 ± 0.90	F = 0.006; *p* = 0.941

**Table 2 jcm-13-02005-t002:** Frequencies of people with Relevant Depressive Symptoms, people with Dysfunctional Rhythms, and people with Relevant Depressive Symptoms plus Dysfunctional Rhythms, from T0 to T1.

	T0	T1	χ^2^	*p*	Yates Correction	*p*	OR (CI95%)
With Dysfunctional Rhythms (DR)	30/93 (32.2%)	23/93 (24.7%)	1.293	0.256	0.9499	0.33	1.45 (0.76–2.75)
With Relevant Depressive Symptoms (RDS)	5/93 (5.4%)	6/93 (6.4%)	0.0966	0.756	0	1	0.82 (0.24–2.8)
With RDS plus DR (% among those with RDS)	4/5 (80%)	4/6 (66.7%)	0.2444	0.621 *	0.034	0.85 *	2 (0.13–31.98)

* Fisher exact test: 1 (not significative).

**Table 3 jcm-13-02005-t003:** Comparisons between people with Relevant Depressive Symptoms (RDS) plus Dysfunctional Rhythms (DR) and the remaining portions of the cohort, at T0 and T1.

	With DR (T0)	Without DR (T0)	Fisher Exact Test	OR (CI95%)
With RDS (T0) N = 5	4 (80%)	1 (20%)		
Without RDS (T0) N = 88	26 (29.6%)	62 (70.4%)		
Total N = 93	30 (32.3%)	63 (67.7%)	*p* = 0.036	9.54 (1.02–89.48)
	With DR (T1)	Without DR (T1)	Fisher exact test	OR (CI95%)
With RDS (T1) N = 6	4 (66.7%)	2 (33.3%)		
Without RDS (T1) N = 87	19 (21.8%)	68 (78.2%)		
Total N = 93	23 (24.7%)	70 (75.3%)	*p* = 0.0313	7.16 (1.22–42.1)

**Table 4 jcm-13-02005-t004:** Comparison between new cases with Relevant Depressive Symptoms (RDS) at T1 with increasing >9 points in BSRS total score from T0 to T1, and the remaining portions of the cohort.

	From T0 to T1 BSRS Total Score > 9 Points (with DR)	From T0 to T1 BSRS Total Score < 10 Points (without DR)	Fisher Exact Test	OR (CI95%)
New Cases with RDS (T1) N = 3	2 (66.7%)	1 (33.3%)		
Not New Cases with RDS (T1) N = 90	5 (5.6%)	85 (94.4%)		
Total N = 93	7 (7.5%)	86 (92.5%)	*p* = 0.014	34.4 (2.6–441.7)

Legend. BSRS: Brief Social Rhythms Scale; DR: Dysfunctional Rhythms.

## Data Availability

The data presented in this study are available on request from the corresponding Author. The data are not publicly available due to privacy and ethical issues.

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
