# Peer review of "The Stability of Social and Behavioral Rhythms and Unexpected Low Rate of Relevant Depressive Symptoms in Old Adults during the COVID-19 Pandemic"

_jcm, 2024, doi:10.3390/jcm13072005_

Round 1
Reviewer 1 Report (Previous Reviewer 2)
Comments and Suggestions for Authors
The manuscript entitled The Stability of Social and Behavioral Rhythms and Unexpected Low Rate of Depression in Old Adults during the COVID-19 Pandemic is presented for the peer review.The disrupted social and circadian rhythms resulting from the stringent lockdown measures have been scrutinized, among other factors, as potential explanations for the heightened prevalence of depressive disorders induced by the COVID-19 pandemic. Several studies suggest that mental health deteriorated in many countries worldwide before and during enforced isolation (ie, lockdown), but it remains unknown how mental health has changed week by week over the course of the COVID-19 pandemic.
I positively assess the improvements that authors have done during resubmission. But still I have some issues to resolve in the submitted manuscript. Please carefully revise your manuscript and put all notes in.
1. Please provide the Italian version at supplements Chapter. Have this version been validated ? I need one more phrase about it in Methods Chapter.
2. Also please provide at least one more method you evaluated daylight duration or light intensity outdoors during test period.
3. Please provide inclusion and exclusion criteria for your study. Moreover, I don't understand your sampling methods.
4. Table 2 misses number of participants. Pleas include statistical signifance in footnotes.
5. The same thing for Tables 3 and 4.
6. Please mention paper by Fancourt doi: 10.1016/S2215-0366(20)30482-X.
Comments on the Quality of English Language
English language is OK.
Author Response
Thank you very much for taking the time to review this manuscript. Please find the detailed responses below and the corresponding revisions/corrections highlighted/in track changes (in red) in the re-submitted manuscript.
Comment 1: Please provide the Italian version at supplements Chapter. Have this version been validated? I need one more phrase about it in Methods Chapter. Response 1: Thank you for this comment. However, it is not clear which Italian versions you are referring to. If you are referring to the two questionnaires (PHQ-9 and BSRS), we have already specified in the text the bibliographic references of both validation studies in Italian. In response to your comment, we have incorporated the following sentence in the Methods section (p. 3, lines 120-121): "A copy of the Italian version of both questionnaires (PHQ-9 and BSRS) is included in the supplementary materials." Comment 2: Also please provide at least one more method you evaluated daylight duration or light intensity outdoors during test period. Response 2: Thank you for this comment. However, daylight duration is not among the outcomes of interest in this study, and therefore it would not be appropriate to specify in the methods section how it can be estimated. Nonetheless, following your observation, we have provided a clearer explanation in the Discussion regarding how we estimated daylight during the observation period, as follows (p. 6, lines: 236-238): "In our study, the first and the follow-up evaluations were performed in April 2019 and April 2020 respectively. In Cagliari (Italy), there were approximately 13 hours of daylight at this time [37], which is slightly above the annual average." Comment 3: Please provide inclusion and exclusion criteria for your study. Moreover, I don't understand your sampling methods Response 3: Thank you for this comment. We better clarified the sampling procedure and provided inclusion and exclusion criteria, as follows (p.2,3, lines 91-106): "A controlled cohort study (secondary analyses) was carried out in Cagliari, Italy on elderly people previously involved in an RCT on exercise for active aging [7,8] and on elderly people recruited for the same RCT, but not enrolled in the study arms due to the overnumbered, which remained in a waiting list for another trial. As reported previously [7,8], the inclusion criteria were: people living at home, 65 years old and older, both genders, holding a medical certificate for non-competitive physical activity. Exclusion criteria were: age ≤ 65 years old, BMI > 35, unsuitability for moderate physical activity due to any medical condition, lifetime history of psychosis and/or mania, a certified organic brain disease, already involved in a program of physical exercise at a level equal or higher than 70% of the expected physical activity level of the intervention study. All people were evaluated for depressive symptoms and rhythms dysfunction at the cohort entry in April 2019 (T0) and 48 weeks after T0, in April 2020 (T1). To assess the association between dysfunctional rhythms and depressive symptoms, the entire cohort was divided into individuals with rhythms dysfunction (BSRS score: above the mean plus one standard deviation) and individuals without rhythms dysfunction, at T0 and T1." Comments 4: Table 2 misses number of participants. Pleas include statistical signifance in footnotes. The same thing for Tables 3 and 4. Response 4: Thank you for this comment. We added the number of participants and split into separate columns the statistical significance in Table 2, 3 and 4. Comments 5: Please mention paper by Fancourt doi: 10.1016/S2215-0366(20)30482-X. Response 5: Thank you for this suggestion. We mentioned the paper in the Introduction Section (p. 2, line 81) and the Discussion section (p. 5, line 184). |
Response to Comments on the Quality of English Language |
Point 1: English language is OK. Response 1: thank you for the approval. |
Reviewer 2 Report (Previous Reviewer 3)
Comments and Suggestions for Authors
Article ``The Stability of Social and Behavioral Rhythms and Unexpected Low Rate of Depression in Old Adults during the
COVID-19 Pandemic.'' has been adequately corrected, it meets all the criteria for publication. The authors adequately answered the reviewers' questions
Author Response
Thank you very much for taking the time to review this manuscript and for your approval. Please find the detailed responses below. Following the comments from the other Reviewers, we highlighted in red all the corrections in the re-submitted manuscript.
Comments 1: Article ``The Stability of Social and Behavioral Rhythms and Unexpected Low Rate of Depression in Old Adults during the COVID-19 Pandemic.'' has been adequately corrected, it meets all the criteria for publication. The authors adequately answered the reviewers' questions. |
Response 1: Thank you for your approval. |
Response to Comments on the Quality of English Language |
Point 1: I am not qualified to assess the quality of English in this paper. |
Reviewer 3 Report (New Reviewer)
Comments and Suggestions for Authors
This study explores the potential interactions between social and behavioral rhythms and the rate of depression in old adults during the COVID-19 pandemic. This is a very interesting topic with potential clinical and therapeutic implications, although the small sample size and lack of control of important variables (e.g., comorbid somatic and/or psychiatric conditions, concomitant pharmacological treatment, lifestyle factors, exposure to artificial light and percentage of time spent interacting with electronic devices/day) preclude any definitive conclusions. Please refer to the additional observations below:
Line 24- „screen for depressive episodes”. Although PHQ-9 has very good specificity and sensitivity for major depressive episodes, only a clinician can validate this diagnosis. Therefore, no reference can be made to the presence of (M)DE, but only to the „depressive symptoms” or the „likelihood of DE” (as well pointed out in line 106, but then unclearly stated in lines 148-150, and not covered in the „Limitations” section)
Lines 90-94- Were the control group controlled for age, gender, baseline depression, circadian rhythms dysfunctions, and other relevant variables? If so, please mention this explicitly.
Line 94- please insert an „=” between „N” and „93”
Line 115- What does „other people” mean in this context? This could refer to individuals with DE without dysfunctional rhythms, individuals with dysfunctional rhythms without DE, both, or individuals with no DE or circadian rhythms dysfunction.
Line 123- „underwent assessment 48 weeks (T1) after the initial evaluation (T0)”
Lines 150-151- formulating statistical conclusions on the presence of circadian rhythm dysfunctions in 3 individuals vs. the remaining sample is questionable due to the extremely small sample
Table 4- the footnote has a typo- „BSRE” instead of „BSRS”
Maybe a short paragraph containing the core ideas, under the name of „Conclusions”, would be useful, in order to highlight the take-home points, and to distinguish them from „Discussions”.
Comments on the Quality of English Language
Minor editing of the English language is needed.
Author Response
Thank you very much for taking the time to review this manuscript. Please find the detailed responses below and the corresponding revisions/corrections highlighted in red in the re-submitted files.
Comment 1: This study explores the potential interactions between social and behavioral rhythms and the rate of depression in old adults during the COVID-19 pandemic. This is a very interesting topic with potential clinical and therapeutic implications, although the small sample size and lack of control of important variables (e.g., comorbid somatic and/or psychiatric conditions, concomitant pharmacological treatment, lifestyle factors, exposure to artificial light and percentage of time spent interacting with electronic devices/day) preclude any definitive conclusions. Response 1: Thank you for these considerations. We have further elaborated the Limitations section as follows (p. 7, Lines 251-259): "The investigation exhibits some limitations. It was carried out with small and unstructured samples, and the high variability in the data could play a role in undermining the statistical power of the findings. Furthermore, assessing only for social/behavioral rhythms without accounting for potential confounding variables (i.e.: exposure to light intensity, both indoors and outdoors; comorbid somatic and/or psychiatric conditions; concomitant pharmacological treatments; lifestyle factors, hours/day spent interacting with electronic devices), precludes any definitive conclusions and does not robustly verify the hypothesis regarding the stability of rhythms, as well as the association between rhythms dysregulation and relevant depressive symptoms." Comment 2: Please refer to the additional observations below. Line 24- „screen for depressive episodes”. Although PHQ-9 has very good specificity and sensitivity for major depressive episodes, only a clinician can validate this diagnosis. Therefore, no reference can be made to the presence of (M)DE, but only to the „depressive symptoms” or the „likelihood of DE” (as well pointed out in line 106, but then unclearly stated in lines 148-150, and not covered in the „Limitations” section). Response 2: Thank you for these observations, which allow us to better clarify this issue in the entire text and the tables, as follows:
Comment 3: Lines 90-94- Were the control group controlled for age, gender, baseline depression, circadian rhythms dysfunctions, and other relevant variables? If so, please mention this explicitly. Response 3: Thank you for posing this question. In response to another Reviewer as well, we have described and further clarified the section dedicated to the design and sample, as follows (pp. 2-3, lines 91-106): "A controlled cohort study (secondary analyses) was carried out in Cagliari, Italy on elderly people previously involved in an RCT on exercise for active aging [7,8] and on elderly people recruited for the same RCT, but not enrolled in the study arms due to the overnumbered, which remained in a waiting list for another trial. As reported previously [7,8], the inclusion criteria were: people living at home, 65 years old and older, both genders, holding a medical certificate for non-competitive physical activity. Exclusion criteria were: age ≤ 65 years old, BMI > 35, unsuitability for moderate physical activity due to any medical condition, lifetime history of psychosis and/or mania, a certified organic brain disease, already involved in a program of physical exercise at a level equal or higher than 70% of the expected physical activity level of the intervention study. All people were evaluated at the cohort entry in April 2019 (T0) and 48 weeks after T0, in April 2020 (T1). To assess the association between dysfunctional rhythms and depressive symptoms, the entire cohort was divided into individuals with rhythm dysfunction (BSRS score: above the mean plus one standard deviation) and individuals without rhythm dysfunction at T0 and T1." Regarding the issue of controlling groups (people with dysfunctional rhythms at T0 vs people without dysfunctional rhythms at T0) concerning possible confounding factors, we would like to further clarify the following: 1) Age: All subjects are aged 65 or older, thus all belonging to the elderly age group. 2) Gender: We have verified and specified in the Methods (p. 3, lines 129-131) and Results (p.5, lines 167-169) sections that at T0, among the 30 subjects with dysfunctional rhythms (risk factor for depression), 10 were males, balanced with the 36 males among the 63 subjects without dysfunctional rhythms (χ² with yates correction=1.1879, p=0.275). 3) Depressive symptoms: we did not control for depressive symptoms at T0. However, the trend regarding the association between rhythm dysfunction as a risk factor for relevant depressive symptoms is confirmed even when we do not consider subjects with relevant depressive symptoms at T0, but only new cases with relevant depressive symptoms at T1 (please see Tables 3 and 4). Comment 4: Line 94- please insert an „=” between „N” and „93” Response 4: thank you for this comment. We corrected as follows (p. 3, line 145): "The total sample (N=93) included:…" Comment 5: Line 115- What does „other people” mean in this context? This could refer to individuals with DE without dysfunctional rhythms, individuals with dysfunctional rhythms without DE, both, or individuals with no DE or circadian rhythms dysfunction. Response 5: Thank you for this question. We have better specified as follows:
Comment 6: Line 123- „underwent assessment 48 weeks (T1) after the initial evaluation (T0)” Response 6: Thank you. We have corrected as indicated (p. 3, line 144) Comment 7: Lines 150-151- formulating statistical conclusions on the presence of circadian rhythm dysfunctions in 3 individuals vs. the remaining sample is questionable due to the extremely small sample. Response 7: Thank you for this observation. In this section about Results, we predominantly presented the data from a descriptive perspective (frequencies and percentages). The comparison was conducted using the Fisher exact test precisely because we accounted for low frequencies. Nevertheless, while this comparison is certainly not exhaustive as underlined in the Limitations section, it indicates a strong trend where, albeit 3 out of 93, the new cases with relevant depressive symptoms at T1 exhibit a higher probability of experiencing an increase of at least 9 points on the BSRS compared to the remaining 90. Finally, we emphasized these aspects also in the Discussion and Conclusions sections, as follows (p. 6, lines 199-205): "Although suggesting only a trend, the findings of the current study further substantiate the association between the exacerbation of rhythm dysregulation and the manifestation of depressive symptoms. Specifically, even if small, the subset of individuals with relevant depressive symptoms (RDS) exhibited a substantial dysregulation in social and behavioral rhythms. Moreover, individuals detected as new cases with RDS which emerged during the lockdown, showed an increase in their BSRS score by >9 points 34 times more than individuals not detected as new cases with RDS." Comment 8: Table 4- the footnote has a typo- „BSRE” instead of „BSRS” Response 8: Thank you for this indication. We corrected the typo (p. 5, line 182) Comment 9: Maybe a short paragraph containing the core ideas, under the name of „Conclusions”, would be useful, in order to highlight the take-home points, and to distinguish them from „Discussions”. Response 9: thank you for this suggestion. We add a section about Conclusions, as follows (p. 7, lines 263-274): "Among a cohort of elderly people who did not show an increased risk of depression during the lockdown prompted by the COVID-19 pandemic, social and behavioral rhythms remained exceptionally stable during the same period. Considering previous evidence about rhythms dysregulation preceding depression, their stability may be considered a factor of resilience in this population. Notably, the unprecedented conditions of the COVID-19 pandemic allowed us just to serendipitously test our hypotheses through a longitudinal study that pointed out interesting results, even if preliminary. They opened the way to the formulation of novel research hypotheses (i.e.: investigating the role of exposure to light pollution in the regulation of social and behavioral rhythms among the elderly) to be further explored by randomized-controlled studies with larger samples." |
Response to Comments on the Quality of English Language |
Point 1: Minor editing of the English language is needed. Response 1: Thank you for your comment. We further revised the English language. |
Round 2
Reviewer 1 Report (Previous Reviewer 2)
Comments and Suggestions for Authors
Authors have addressed all my issues. Thank you
Author Response
Thank you very much for taking the time to review this manuscript. Please find the detailed responses below and the corresponding revisions/corrections highlighted in blue in the re-submitted file.
Point-by-point response to Comments and Suggestions for Authors
Comment 1: Authors have addressed all my issues. Thank you
Response 1: Thank you.
Reviewer 3 Report (New Reviewer)
Comments and Suggestions for Authors
The manuscript significantly improved due to the diligent work of the Authors. Please refer to the comments below:
Line 21- Each abbreviation should be explained the first time they are used, even obvious ones, like RCT; however, if an abbreviation is used only one time, as is the case of RCT in the „Abstract” section, then consider using the spelled-out version of that concept only;
Line 109- Any Cronbach alpha values for these tools?
Line 120- the inclusion of PHQ-9 and BSRS Italian versions in the Supplementary materials section is most welcome.
Author Response
Thank you very much for taking the time to review this manuscript. Please find the detailed responses below and the corresponding revisions/corrections highlighted in blue in the re-submitted file.
Point-by-point response to Comments and Suggestions for Authors
Comments 1: The manuscript significantly improved due to the diligent work of the Authors.
Response 1: thank you for your appreciation.
Comment 2: Please refer to the comments below:
Line 21- Each abbreviation should be explained the first time they are used, even obvious ones, like RCT; however, if an abbreviation is used only one time, as is the case of RCT in the „Abstract” section, then consider using the spelled-out version of that concept only
Response 2: Thank you for this indication. We have corrected as requested in line 21 of the Abstract and reviewed the entire text.
Comment 3: Line 109- Any Cronbach alpha values for these tools?
Response 3: Thank you for this observation. We add the Cronbach alpha value for the Italian validation of BSRS (p. 3, line 113) and for the Italian version of PHQ-9 (p.3, line 119) as follows:
Cronbach’s alpha coefficient was 0.912 [28].
Cronbach’s alpha coefficient was 0.79 [29].
Comment 4: Line 120- the inclusion of PHQ-9 and BSRS Italian versions in the Supplementary materials section is most welcome.
Response 4: Thank you for your appreciation.
This manuscript is a resubmission of an earlier submission. The following is a list of the peer review reports and author responses from that submission.
Round 1
Reviewer 1 Report
Comments and Suggestions for Authors
- The Title suggests a study on the role of light pollution in relation to depression and social/behavioral rhythms during COVID-19, but the introduction, methods, and results do not align with this. There is a lack of measures or results related to light exposure. In fact, I only see light mentioned in the discussion.
- Utilizing only one scale for depression and social/behavioral rhythms without accounting for confounding variables does not robustly verify the stability of social rhythms. This method does not meet the standard for establishing a reliable relationship between the variables.
- This paper does not mention the study design, which I suspect to be a secondary analysis of data from a parent RCT.
- The extremely wide confidence interval in Table 3 (e.g., 1.02-89.48) raises concerns about the statistical power and precision of the results. Wide confidence intervals can indicate a small sample size or high variability in the data, which undermines the reliability of the findings.
Comments on the Quality of English Language
Acceptable
Author Response
Please, see the attachment. The last file is the correct one.

Reviewer 2 Report
Comments and Suggestions for Authors
Authors presented the manuscript entitled Stability of Social and Behavioral Rhythms and Unexpected Low Rate of Depression in Old Adults during the COVID-19 Pandemic: Could Light Pollution Play a Role? for the peer review. The paper is of great interest, especially for geriatrics, medical professionals and scientists. COVID19 viral outbreak had great impact on isolation and activity restriction in elderly people. The disruption of social rhythms was found associated with depressive disorders during the COVID-19 pandemic, when lower rates of these disorders were surprisingly found in old adults.
Some issues to be fixed before acceptance.
1. Authors postulate that depression dysregulates circadian rhythms in people. Please point out direct circadian markers you have measured in the study.
2. Have you measured light intensity both outdoors and indoors? Light plays pivotal role in depression development.
3. Please mark time of the year you performed the study. Also, specify daylight duration at these time points.
4. Have you meant light pollution as artificial light ar night? It is not clear from Introduction.
Author Response
Please, see the attachment.

Reviewer 3 Report
Comments and Suggestions for Authors
Article
,,Stability of Social and Behavioral Rhythms and Unexpected 2
Low Rate of Depression in Old Adults during the COVID-19 3
Pandemic: Could Light Pollution Play a Role? ,, is a well-written, methodologically well-placed article with an adequate goal and conclusion. The introductory part discusses the possibility of air pollution and daily circadian rhythms, that is, light on the psychological problems of adults. Mood disorders are closely associated with the dysregulation of biological and social rhythms (sleeping, eating, carrying out social activities), which are, in turn, directly related to circadian biorhythms. The disrupted social and circadian rhythms resulting from the stringent lockdown measures have been scrutinized, among other factors, as potential explanations for the heightened prevalence of depressive disorders induced by the COVID-19 pandemic. It has been shown that people who spend more time in closed spaces suffer from some forms of psychosomatic and psychosocial disorders that can be reflected, depending on the age group and predictive parameters, even in severe forms of depression and even suicides.The discussion is adequately accompanied by references and the results are clearly presented.This could be attributed to their retirement status, which may imply less engagement in remote work, or cultural preferences leading to less use of e-tablets and computers. Other factors may be considered or implicated as contributing causes to the emergence of the paradox in which the rate of depressive symptoms did not increase during the lockdown in the elderly. For individuals with work and school obligations, the lockdown might have posed more challenges in terms of adaptation compared to retired elderly individuals, whose days were characterized by fewer hectic extra-domestic commitments. Moreover, in the 207 countries examined in the research regarding the reduced risk of depression in the elderly
during the COVID-19 pandemic (USA, Canada, Italy), the majority of older individuals received pension benefits. Unlike workers in commercial or business sectors, whose incomes were at risk, the elderly were less exposed to economic uncertainty during the pandemic. Hence, it is plausible that the elderly face less economic uncertainty. Nevertheless, the unexpected stability of social and behavioral rhythms identified by our study in the elderly during the COVID-19 pandemic has proven to be closely related to the low risk of depression in old adults, as later confirmed by several studies. Therefore, the data needs validation and additional exploration, as does the hypothesis of a central role in lower exposure to light pollution. The investigation exhibits a limitation as it was carried out with a small and unstructured sample for testing the hypothesis underlying this specific study. Nevertheless, the 219
perspective that allowed us to verify the hypothesis serendipitously is, however, a condition that is unrepeatable in some respects, and this enhances these results.
excellently written epidemiologic study, clinically adequately good results, which would serve to better organize people during the next trials during pandemics/epidemics
​
Author Response
Please, see the attachment.

Round 2
Reviewer 2 Report
Comments and Suggestions for Authors
I think authors could perform simple experiments in different light-dark conditions for senior citizens and made objective control tools, too.
Lack of necessary experiments and control group is crucial in my decision.
Author Response
Thank you for your comment. However, it refers to the topic of light pollution which, in the present re-elaboration, has become a marginal aspect and only hypothesized as a heuristic hypothesis. It is suggested as a topic to be underlined in future studies, as we better specified as follows (page 7, lines: 225-231):
The investigation exhibits some limitations. It was carried out with a small and unstructured sample. Nevertheless, the unprecedented conditions of the COVID-19 pandemic allowed us to serendipitously test our hypothesis only through a longitudinal study. This methodology enabled us to obtain interesting results, even if preliminary, opening the way to the formulation of novel research hypotheses (i.e.: investigating the role of exposure to light pollution in the regulation of social and behavioral rhythms among the elderly) to be further explored through randomized-controlled studies with larger samples.
We changed the Title as follows (page 1, lines:1-4):
The Stability of Social and Behavioral Rhythms and Unexpected Low Rate of Depression in Old Adults during the COVID-19 Pandemic.
We finally removed from the Abstract the following statement:
One possible explanation could be the lower exposure to artificial light from computers and tablets during the waking day because old adults were not involved in remote work and less involved in the documented exponential increase in play games during periods of lockdown.
Please find the corresponding revisions/corrections highlighted/in track changes in red (Round 1 – reviewing process) and in blue (Round 2 – reviewing process) in the re-submitted manuscript.
